# Masked Distillation Advances Self-Supervised Transformer Architecture Search

**Caixia Yan**
School of Computer Science and Technology
MOEKLINNS Laboratory, Xi'an Jiaotong University
`yancaixia@xjtu.edu.cn`

**Xiaojun Chang**
University of Science and Technology of China
Mohamed bin Zayed University of Artificial Intelligence
`cxj273@gmail.com`

**Zhihui Li**
School of Information Science and Technology
University of Science and Technology of China
`zhihuilics@gmail.com`

**Lina Yao**
CSIRO's Data61
University of New South Wales
`lina.yao@data61.csiro.au`

**Minnan Luo**
School of Computer Science and Technology
MOEKLINNS Laboratory, Xi'an Jiaotong University
`minnluo@xjtu.edu.cn`

**Qinghua Zheng** *
School of Computer Science and Technology
MOEKLINNS Laboratory, Xi'an Jiaotong University
`qhzheng@mail.xjtu.edu.cn`

## Abstract

Transformer architecture search (TAS) has achieved remarkable progress in automating the neural architecture design process of vision transformers. Recent TAS advancements have discovered outstanding transformer architectures while saving tremendous labor from human experts. However, it is still cumbersome to deploy these methods in real-world applications due to the expensive costs of data labeling under the supervised learning paradigm. To this end, this paper proposes a masked image modelling (MIM) based self-supervised neural architecture search method specifically designed for vision transformers, termed as MaskTAS, which completely avoids the expensive costs of data labeling inherited from supervised learning. Based on the one-shot NAS framework, MaskTAS requires to train various weight-sharing subnets, which can easily diverged without strong supervision in MIM-based self-supervised learning. For this issue, we design the search space of MaskTAS as a siamesed teacher-student architecture to distill knowledge from pre-trained networks, allowing for efficient training of the transformer supernet. To achieve self-supervised transformer architecture search, we further design a novel unsupervised evaluation metric for the evolutionary search algorithm, where each candidate of the student branch is rated by measuring its consistency with the larger teacher network. Extensive experiments demonstrate that the searched architectures can achieve state-of-the-art accuracy on CIFAR-10, CIFAR-100, and ImageNet datasets even without using manual labels. Moreover, the proposed MaskTAS can generalize well to various data domains and tasks by searching specialized transformer architectures in self-supervised manner.

## 1 Introduction

Vision transformer (ViT), as a self-attention characterized neural network, has recently emerged as an alternative to convolutional neural networks (CNNs) in computer vision community. In an attempt to progress this field, a surge of advanced ViT networks have been developed over a wide range of vision tasks, such as image recognition Dosovitskiy et al. (2020); Han et al. (2021); Liu et al. (2021b), object detection Liu et al. (2021b); Zhu et al. (2020) and semantic segmentation Strudel et al. (2021); Xie et al. (2021). However, despite the empirical success of these approaches, they all

---

*Corresponding Author

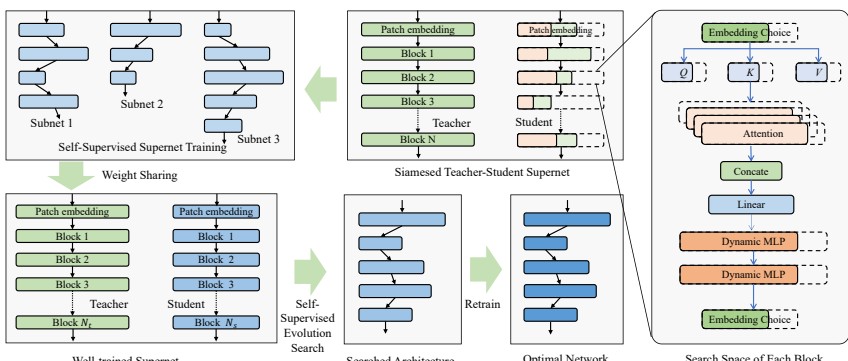

Figure 1: The pipeline of the proposed self-supervised transformer architecture search framework, including a) self-supervised supernet training, b) self-supervised architecture search, and c) supervised re-training of searched architecture.

rely heavily on hand-crafted ViT architectures designed by human experts, meaning that laborious trial-and-error testing is required. In particular, a large number of design choices needs to be analyzed and determined in ViT architecture, such as patch size of input, number of self-attention heads, query/key/value dimension, MLP ratio and network depth, significantly increasing the difficulty of architecture design. Moreover, these manually designed architectures are fixed once obtained, which cannot necessarily ensure the optimality for diversified data domains and task scenarios.

Neural Architecture Search (NAS) Ren et al. (2021); Liu et al. (2021a); Elsken et al. (2019); Yan et al. (2021) has become an effective way for automating the design of neural networks. Inspired by its remarkable success in CNNs, NAS techniques have also been investigated for searching standard transformer architectures recently. Existing studies on Transformer Architecture Search (TAS) mainly focus on refining search space (*e.g.*, S3 Chen et al. (2021c), GLiT Chen et al. (2021a) and AutoFormer Chen et al. (2021b)) and/or improving search algorithms (*e.g.*, AutoFormer Chen et al. (2021b) and ViTAS Su et al. (2021)). Benefiting from the automated network design process, these methods can discover customized architectures that can achieve comparable or better performance, compared with manually designed vision transformers. However, despite their empirical success, the training and architecture search stage of these approaches are both performed in a fully-supervised paradigm, meaning that the optimal transformer architectures are typically searched based on both images and their associated labels. This dependency would inevitably lead to expensive costs of data labelling, limiting the scalability and transferability of the transformer architecture search approaches. As a viable alternative, self-supervised learning obtains supervisory signals from the data itself and has recently been shown to address the appetite for data successfully Chen et al. (2020); Grill et al. (2020); Wang et al. (2021). A few recent works have explored how to discover desirable network architectures based on self-supervised search strategy. However, they are concentrated on convolutional modules in CNNs, which are quite different from vision transformers.

In this paper, we firstly explore self-supervised neural architecture search specifically for vision transformers, dedicated to search promising vision transformer architectures in a self-supervised manner. Concretely, we take advantage of the efficacy of Masked Image Modeling (MIM) Wei et al. (2022); He et al. (2022); Feichtenhofer et al. (2022); Assran et al. (2022) in developing scalable ViT models, and propose a MIM based self-supervised ViT architecture search method, termed as MaskTAS. Following the standard one-shot NAS pipeline, the overall framework of MaskTAS can be decomposed into two stages, *i.e.*, supernet training and subnet search. The overall pipeline of the proposed MaskTAS has been demonstrated in Figure 1.

To enable self-supervised supernet training, the proposed MaskTAS adapts the standard one-shot NAS framework to the regime of MIM based self-supervised learning. During supernet training, millions of weight-sharing subnets in the supernet need to be trained by masking and recovering random image patches. However, due to the lack of strong and stable supervision, the co-training of various subnets is prone to diverging, leading to suboptimal results. To this end, we build the MaskTAS supernet as a teacher-student based siamese architecture, where the pre-trained larger teacher network can provide strong supervision to facilitate the efficient training of student branch. During supernet training stage, each sampled architecture of the student network can be updated by 1) reconstructing the original image from the masked signals, and 2) predicting the feature of the teacher branch, in a self-supervised manner.

After the supernet training is complete, the architecture search stage can be performed by rating the candidates sampled from the well-optimized supernet. To avoid the dependency on image labels, we design an unsupervised evaluation metric for the evolutionary search algorithm, where the performance of each candidate architecture of the student is measured by computing its feature similarity with the teacher network. In this way, the teacher with stabilized behavior can act as the evaluation target for the sampled architectures of the student branch. Once the optimal architecture has been found, it can be retrained on the target dataset after inheriting the weights from the well-optimized supernet. The contributions of the present work can be summarized as follows:

- We establish a MIM-based self-supevised learning framework, called MaskTAS, to pursue scalable and transferable ViT architecture search. To the best of our knowledge, this is the earliest effort to develop self-supervised architecture search paradigm for ViTs.
- We reveal the key challenge in self-supervised supernet training, *i.e.*, the diverging of diverse subnets co-training. A siamesed teacher-student supernet is designed to address this challenge by introducing extra supervision from a pre-trained teacher network.
- To allow for self-supervised architecture search, we further design a self-supervised evaluation metric based evolutionary search algorithm, where each candidate of the student branch is rated by measuring its consistency with the teacher network.

## 2 THE PROPOSED METHODOLOGY

In this section, we first briefly review existing one-shot NAS on vision Transformers as preliminary and further point out the dilemma of those methods on producing scalable transformer architectures. Then we present the proposed MaskTAS to address this issue, along with its two key components: i) self-supervised supernet training with masked feature distillation, and ii) evolutionary architecture search based on unsupervised metric.

### 2.1 REVISITING ONE-SHOT NAS TOWARDS TRANSFORMERS

Given a pre-defined transformer search space $\mathcal{S}$, one-shot NAS typically encodes $\mathcal{S}$ into a weight-sharing supernet $\mathcal{N}(\mathcal{S}, W_{\mathcal{S}})$ with weights $W_{\mathcal{S}}$. The supernet includes various weight-sharing subnet architectures as candidates, denoted as $\{\alpha_i\}_{i=1}^N$, where $N$ refers to the total number of candidates with $\alpha_i$ being the $i$-th candidate. Without loss of generality, the search of optimal architecture $\alpha^*$ can be formulated as a two-stage optimization problem. The first stage aims to optimize the supernet weights $W_{\mathcal{S}}$ by solving the following problem:

$$W_{\mathcal{S}}^* = \arg \min_{W_{\mathcal{S}}} \mathcal{L}_{\text{train}}(\mathcal{N}(\mathcal{S}, W_{\mathcal{S}}), \mathcal{D}_{\text{train}}), \tag{1}$$

where $\mathcal{L}_{\text{train}}(\cdot)$ refers to the loss on the training dataset $\mathcal{D}_{\text{train}}$; $W_{\mathcal{S}}^*$ represents the optimal weights after optimization. Following the weight-sharing strategy Li et al. (2021); Chen et al. (2021b), existing one-shot NAS methods sample and train different subnet architecture paths in each training iteration, so as to avoid exhausted memory usage.

In the second stage, the supernet $\mathcal{N}(\mathcal{S}, W_{\mathcal{S}})$ with learned optimal weights $W_{\mathcal{S}}^*$ is directly utilized for subnet architecture search. The optimal architecture $\alpha^*$ can be searched by ranking the performance of subnets with the weights inherited from $W_{\mathcal{S}}^*$, *i.e.*,

$$\alpha^* = \arg \max_{\alpha \in \mathcal{S}} \text{Acc}_{\text{val}}(\mathcal{N}(\alpha, W_{\alpha}^*), \mathcal{D}_{\text{val}}),$$
$$s.t. \quad g(\mathcal{N}(\alpha, W_{\alpha}^*)) \leq \mathcal{C}, \tag{2}$$

where $\text{Acc}_{\text{val}}(\cdot)$ denotes the top-1 accuracy of the subnet $\alpha$ over the validation dataset $\mathcal{D}_{\text{val}}$; $W_{\alpha}^*$ refers to the weights of $\alpha$ inherited from $\mathcal{N}(\mathcal{S}, W_{\mathcal{S}})$. The function $g(\cdot)$ calculates the resource consumption of each model architecture with $\mathcal{C}$ being the given resource constraint.

A few recent works have explored the one-shot NAS framework for vision transformer architecture search, including AutoFormer Chen et al. (2021b), ViTAS Su et al. (2022) and NASformer Ni et al. (2022). Despite their empirical success, these methods are limited to perform training and searching in the regime of supervised learning, which require tremendous labeled images for training to guarantee the searching of desirable architectures. Moreover, they rely heavily on the domain and quality of data annotation, resulting in insufficient generalization ability of the searched architectures.

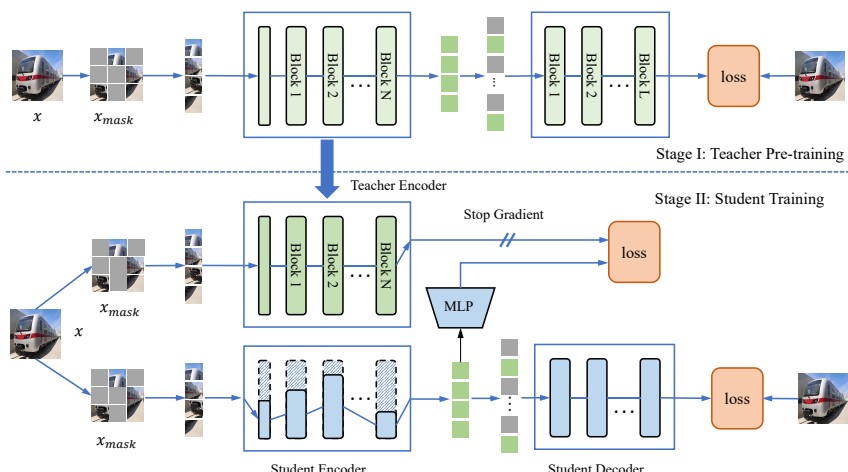

Figure 2: The self-supervised supernet training of the proposed MaskTAS, which performs: (a) teacher pre-training with masked pixel regression and (b) student training with masked feature distillation, in a self-supervised manner.

## 2.2 OVERVIEW OF THE PROPOSED MASKTAS

In order to avoid the dependency on manual annotation, we develop a self-supervised neural architecture search method for transformers by replacing the supervised signal with a self-supervised counterpart. The overview of MaskTAS is presented in Figure 2.

---

**Algorithm 1** Self-supervised Supernet Training in MaskTAS.

---

**Require:** Search space $\mathcal{A}$; Teacher network $\mathcal{T}$; Student network $\mathcal{S}$; Training set $\mathcal{D}_{tr}$; Loop number in an epoch $N_{iter}$; Loss $\mathcal{L}_{train}^t$ and $\mathcal{L}_{train}^s$; max epochs $N_e$.

1: **Initialize student** $\mathcal{N}(\mathcal{S}, \Theta_{\mathcal{S}})$ with weights $\Theta_{\mathcal{S}}$;
2: **Initialize teacher** $\mathcal{N}(\mathcal{T}, \Omega_{\mathcal{T}}^*)$**:** Pre-Train the teacher network $\mathcal{T}$ with $\mathcal{L}_{train}^t$ to get optimal weights $\Omega_{\mathcal{T}}^*$;
3: **for** epoch = 1 to $N_e$ **do**
4:     **for** $i = 1, 2, \cdots, N_{iter}$ **do**
5:         Sample a mini-batch of image data $I^i$ from $\mathcal{D}_{tr}$;
6:         Random masking on $I^i$ to get the masked data $X_v^i$;
7:         Sample a subnet architecture $\mathcal{A}^{(i)}$ with weights $\Theta_{\mathcal{A}^{(i)}}$ from the student branch of supernet;
8:         Feed $X_v^i$ into the teacher-student supernet and compute gradients by $\nabla_{\Theta_{\mathcal{A}^{(i)}}} \mathcal{L}_{train}^s$;
9:         Update subnet weights $\Theta_{\mathcal{A}^{(i)}}$ while freezing the rest parameters of supernet: $\Theta_{\mathcal{A}^{(i)}} \leftarrow$ Adam$(\nabla_{\Theta_{\mathcal{A}^{(i)}}} \mathcal{L}_{train}^s, \Theta_{\mathcal{A}^{(i)}})$;
10:     **end for**
11: **end for**
**Ensure:** Supernet architecture with optimal weights $\mathcal{N}(\mathcal{S}, \Theta_{\mathcal{S}}^*)$ and $\mathcal{N}(\mathcal{T}, \Omega_{\mathcal{T}}^*)$.

---

**Model Overview** The proposed MaskTAS is designed as a siamesed network architecture, which consists of two neural networks that interact with each other, *i.e.*, teacher and student network. As shown in Figure 2, the teacher network parameterized by $\Omega$ is implemented as a transformer encoder $\mathcal{E}_\Omega$ and decoder $\mathcal{R}_\Omega$, while the student network equipped with parameters $\Theta$ includes a transformer encoder $\mathcal{E}_\Theta$ and a decoder $\mathcal{R}_\Theta$. Besides, a feature predictor $\mathcal{M}$ is further introduced to align the output dimension of the teacher and student network.

To explore the possibility of optimal ViT in architecture-level, the architecture of student encoder $\mathcal{E}_\Theta$ is designed as a transformer supernet $\mathcal{N}(\mathcal{A}, \Theta_{\mathcal{A}})$. The search space $\mathcal{A}$ contains all possible architectures that can be searched for transformers. For the purpose of flexible architecture search, we incorporate all the essential factors in ViTs to formulate the model search space, including *patch embedding dimension*, *number of heads*, *MLP ratio* and *depth of architecture*. The supernet contains various weight-sharing subnet architectures, which can be formulated as

$$\mathcal{N}(\mathcal{A}, \Theta_{\mathcal{A}}) = \{\mathcal{N}(\mathcal{A}^{(i)}, \Theta_{\mathcal{A}^{(i)}})\}_{i=1}^K, \tag{3}$$

where $\Theta_{\mathcal{A}^{(i)}}$ refers to the weights of the $i$-th subnet $\mathcal{A}^{(i)}$; $K$ denotes the total number of subnets. We introduce three types of supernets for the student branch, *i.e.*, supernet-tiny, supernet-small and supernet-base, each of which corresponds to different range of parameters to satisfy different resource constraints.

Based on the siamesed supernet architecture defined above, the proposed MaskTAS performs architecture search through a two-step pipeline, including self-supervised supernet training and architecture search.

## 2.3 SELF-SUPERVISED SUPERNET TRAINING

In this section, we formulate the training of supernet $\mathcal{N}(\mathcal{A}, \Theta_{\mathcal{A}})$ as a process of masked distillation based self-supervised learning. As illustrated in Figure 2, our supernet training process is accomplished by two stages, *i.e.*, teacher pre-training and student training. Specifically, we present the overall procedure for self-supervised supernet training in Algorithm 1.

**Teacher Pre-training** Driven by the success of the masked auto-encoder in self-supervised pre-training, we perform masked image modelling to optimize the teacher network parameters $\Theta$ underpinned by pixel reconstruction loss. The objective of pixel reconstruction is to recover the original image from the masked one with an encoder-decoder scheme.

Given an input image $I \in \mathbb{R}^{H \times W \times C}$, we first divide it into $N$ non-overlapping patches of size $P^2 C$ pixels, where $H$, $W$, $C$ refer to the height, width, channel of the image respectively. Thus, each image can be represented by a set of flatted patch vectors $X = \{x_1, x_2, \cdots, x_N\}$. Following MAE He et al. (2022), we uniformly sample a subset of patches from $X$ and mask the remaining ones. This results in two non-overlapping patch sets $X_v = \{x_1^v, x_2^v, \cdots, x_{N_v}^v\}$ and $X_m = \{x_1^m, x_2^m, \cdots, x_{N_m}^m\}$, where $X_v$ and $X_m$ contain $N_v$ visible and $N_m$ masked patches respectively. The masked signal $X_v$ is fed into the encoder of teacher network to generate the latent representation:

$$Z_v = \mathcal{E}_\Omega(X_v | \mathcal{T}_e, \Omega_{\mathcal{T}_e}), \tag{4}$$

where $Z_v$ refers to the latent representation of the masked image and $\mathcal{T}_e$ denotes the architecture of teacher encoder with parameters $\Omega_{\mathcal{T}_e}$. Then, $Z_v$ is integrated with the masked tokens to generate a full set of signals $Z_v'$, which is further fed into the teacher decoder, *i.e.*,

$$P = \mathcal{R}_\Omega(Z_v' | \mathcal{T}_d, \Omega_{\mathcal{T}_d}), \tag{5}$$

where $\mathcal{T}_d$ and $\Omega_{\mathcal{T}_d}$ correspond to the architecture and parameters of the decoder. Finally, the teacher network can be optimized by minimizing the pixel-level MSE loss between the predicted targets $P$ and the original image $I$ over the masked patches.

**Student Training** Under the one-shot NAS framework, the training of student supernet involves the co-training of diverse subnets, which trains one subnet architecture $\mathcal{A}^{(i)}$ from $\mathcal{A}$ in each training iteration. Each subnet $\mathcal{A}^{(i)}$ is uniformly sampled from $K$ candidates, *i.e.*,

$$\begin{cases} \mathcal{A}^{(i)} \in \{\mathcal{A}^{(1)}, \mathcal{A}^{(2)}, \cdots, \mathcal{A}^{(k)}, \cdots, \mathcal{A}^{(K)}\}, \\ \Theta_{\mathcal{A}^{(i)}} \in \{\Theta_{\mathcal{A}^{(1)}}, \Theta_{\in^{(i)}}, \cdots, \Theta_{\mathcal{A}^{(k)}}, \cdots, \Theta_{\mathcal{A}^{(K)}}\}, \end{cases} \tag{6}$$

where $\mathcal{A}^{(i)}$ and $\Theta_{\mathcal{A}^{(i)}}$ denote the architecture and weights of the $i$-th candidate respectively. The architecture of the encoder is dynamically changing during the self-supervised training process. However, without strong supervision signals, the joint training of various weight-sharing networks is prone to diverging, leading to thousands of epochs for pre-training.

To enable efficient training of diverse subnets, MaskTAS further employs knowledge distillation technology Ren et al. (2023); Hinton et al. (2015) to transfer the knowledge of larger MIM pre-trained teacher models to smaller students. First, the masked signal $X_v$ is fed into the encoder of both the teacher and student branch to output the latent representation:

$$\begin{cases} Z_t = \mathcal{E}_\Omega(X_v | \mathcal{T}_e, \Omega_{\mathcal{T}_e}^*), \\ Z_s = \mathcal{E}_\Theta(X_v | \mathcal{A}^{(i)}, \Theta_{\mathcal{A}^{(i)}}), \end{cases} \tag{7}$$

where $\Omega_{\mathcal{T}_e}^*$ is the pre-trained weights of teacher encoder; $Z_t$ and $Z_s$ refer to the latent representation generated by the teacher and student network respectively. The objective is to train the randomly

sampled student subnets efficiently by mimicking the target produced by the teacher in a knowledge distillation manner.

In general, the feature dimensions of the teacher network and the student network are mismatched. To tackle this problem, we adopt an extra projection network $\mathcal{M}$ on the output of the student network to match the feature dimension of the teacher's target. Thus, the latent representation produced by the student encoder is forwarded to predict: 1) the high-level feature of the teacher branch, and 2) the original image in pixel-level:

$$
\begin{cases}
\tilde{Z}_s = \mathcal{M}(Z_s | \sigma, W_\sigma), \\
\tilde{P} = \mathcal{R}_\Theta(Z'_s | \mathcal{S}_d, \Theta_{\mathcal{S}_d}),
\end{cases} \tag{8}
$$

where $\mathcal{S}_d$ denotes the student decoder; $\sigma$ is the architecture of projection network $\mathcal{M}$ with weights $W_\sigma$; $Z'_s$ denotes the full signal derived from $Z_s$; $\tilde{Z}_s$ and $\tilde{P}$ refer to the predicted feature-level and pixel-level information respectively. During training, the architecture of the student branch varies by uniformly sampling, while the teacher branch is fixed to the pre-trained weights for stabilized behavior. After pre-training, the well-optimized supernet can be transferred to various downstream tasks.

**Self-Supervised Training Objective** The supernet is trained in a self-supervised manner by 1) reconstructing the original image after the masking operation, and 2) predicting the high-level feature produced by the teacher network. Given an image $I$ as input, the overall objective function for supernet training can be derived as

$$
\begin{cases}
\mathcal{L}^t_{train} = \mathcal{L}_{reg}(P, I), \\
\mathcal{L}^s_{train} = \mathcal{L}_{reg}(\tilde{P}, X_m) + \beta \mathcal{L}_{pre}(\tilde{Z}_s, Z_t),
\end{cases} \tag{9}
$$

where $\mathcal{L}^t_{train}$ and $\mathcal{L}^s_{train}$ correspond to the loss for teacher pre-training and student training respectively; $\mathcal{L}_{reg}$ and $\mathcal{L}_{pre}$ denote the loss for pixel regression and feature prediction respectively with trade-off parameter $\beta$. More specifically, $\mathcal{L}_{reg}$ computes the $\ell_2$-based reconstruction loss between the original image and the reconstructed image over masked patches, *i.e.*,

$$
\mathcal{L}_{reg}(\tilde{P}, X_m) = \frac{1}{N_m} \sum_{k=1}^{N_m} \frac{1}{P^2 C} \| \tilde{p}_k - \overline{x}_k^m \|_2^2, \tag{10}
$$

where $\overline{x}_k^m$ refers to the normalized patch of $x_k^m \in X_m$ based on the mean and standard deviation; $\tilde{p}_k$ is the $k$-th element of $\tilde{P}$; $N_m$ stands for the number of masked patches. Besides, $\mathcal{L}_{pre}$ is the feature prediction loss used for knowledge distillation, which can be formulated as:

$$
\mathcal{L}_{pre}(\tilde{Z}_s, Z_t) = \text{Smooth}_{\ell_1}(LN(\tilde{Z}_s), LN(Z_t)), \tag{11}
$$

where $LN(\cdot)$ refers to the layer normalization operation and $\text{Smooth}_{\ell_1}(\cdot)$ denotes the smooth $\ell_1$ loss.

## 2.4 Self-Supervised Architecture Search

Once the supernet is trained to converge, we can perform architecture search to find the optimal subnets from the well-optimized student encoder $\mathcal{N}(\mathcal{A}, \Theta^*_\mathcal{A})$. To allow for self-supervised architecture search, we design a teacher-student consistency based evaluation metric, thus adapting the standard evolutionary search algorithm Guo et al. (2020) to an unsupervised paradigm.

**Evolutionary Search** Without loss of generality, the evolutionary search process begins with the generation of $N_r$ random architectures as initialized population. In each iteration, the architectures with top-$k$ performance are selected as parents in each generation, which are further utilized to generate the next generation through crossover and mutation. For a crossover, two randomly selected candidates are picked and crossed to produce a new one during each generation. For mutation, a candidate mutates its depth with probability $P_d$ first. Then it mutates each block with a probability of $P_m$ to produce a new architecture. The optimal subnet architecture can be discovered after a sufficient number of iterations:

$$
\alpha^* = \arg\max_{\alpha \in \mathcal{A}} \text{Acc}(\mathcal{N}(\alpha, \Theta^*_\alpha), \mathcal{D}_{tar}), \quad s.t. \quad g(\mathcal{N}(\alpha, \Theta^*_\alpha)) \leq \mathcal{C}, \tag{12}
$$

where $\mathcal{D}_{tar}$ stands for the target dataset; $\alpha$ is the sampled subnet architecture from $\mathcal{A}$; $\Theta^*_\alpha$ refers to the weights of subnet $\alpha$ inherited from $\Theta^*$; $\text{Acc}(\cdot)$ is the performance evaluation metric for candidate architectures.

**Self-supervised Architecture Evaluation** Driven by the universally acknowledged principle that the larger the better works for deep neural networks, we believe the large teacher network with pre-trained weights can provide a fair rating for each sampled architecture of the student branch. Each candidate architecture of the student-branch can be rated by computing its feature similarity with the teacher-branch, *i.e.*,

$$\text{Acc}(\mathcal{N}(\alpha, \Theta_\alpha^*), \mathcal{D}_{tar}) = \sum_{i=1}^{|\mathcal{D}_{tar}|} \mathcal{H}(Z_t^i, Z_s^i), \tag{13}$$

where

$$\begin{cases} Z_t^i = \mathcal{E}_{\Omega^*}(X_v^i|\mathcal{T}_e, \Omega_{\mathcal{T}_e}^*), \\ Z_s^i = \mathcal{E}_{\Theta^*}(X_v^i|\alpha, \Theta_\alpha^*), \end{cases} \tag{14}$$

where $\mathcal{H}(\cdot)$ is the similarity function; $X_v^i$ refers to the visible patch set after random masking of the $i$-th image; $Z_t^i$ and $Z_s^i$ refer to the output feature of the teacher and student encoder respectively.

The similarity of two feature maps, *i.e.*, $Z_t^i$ and $Z_s^i$, cannot be directly measured, since the output dimensions are different for the teacher and student encoder. Thus, we propose to evaluate the similarity of two feature maps by computing the relative relation of features, *i.e.*,

$$\mathcal{H}(Z_t^i, Z_s^i) = -\sum_{j=1}^{N_v} \sum_{k=1}^{N_v} h_{jk}^t \log h_{jk}^s, \tag{15}$$

where

$$\begin{cases} h_{jk}^t = -\log \frac{\exp(Z_t^{i,j} Z_t^{i,k}/\tau)}{\sum_{k=1}^{N_m} \exp(Z_t^{i,j} Z_t^{i,k}/\tau)}, \\ h_{jk}^s = -\log \frac{\exp(Z_s^{i,j} Z_s^{i,k}/\tau)}{\sum_{k=1}^{N_m} \exp(Z_s^{i,j} Z_s^{i,k}/\tau)}, \end{cases} \tag{16}$$

where $Z_t^{i,j}$ and $Z_s^{i,j}$ are the $j$-th row of $Z_t^i$ and $Z_s^i$ respectively; $h_{jk}^t$ and $h_{jk}^s$ denote the relative relation between the $j$-th and $k$-th row of $Z_t^i$ and $Z_s^i$ respectively.

## 3 EXPERIMENTS

In this section, we first provide the implementation details of the proposed MaskTAS. Then we present the performance of MaskTAS with comparisons to other state-of-the-art models designed manually or automatically. Finally, we conduct ablation studies to further evaluate the effectiveness of MaskTAS.

### 3.1 EXPERIMENTAL SETUP

**Dataset Description.** We evaluate the effectiveness of the proposed method over the large-scale ImageNet Russakovsky et al. (2015) dataset, which contains 1.28 million images in 1000 categories collected for the image classification task. We also experiment on other classification tasks, including CIFAR-10 Krizhevsky et al. (2009), CIFAR-100 Krizhevsky et al. (2009), PETS Parkhi et al. (2012) and Flowers Nilsback & Zisserman (2008), to evaluate the transferability of the self-supervised architecture search method. More specifically, the CIFAR-10 dataset consists of 60000 images in 10 classes, with 6000 images per class. The CIFAR-100 dataset has 100 classes containing 600 images each. PETS is a 37 category pet dataset with roughly 200 images for each class. The Flowers dataset consists of 102 flower categories, where each class consists of between 40 and 258 images. Moreover, the transferability of the searched architectures is verified by ADE20K Zhou et al. (2019) semantic segmentation task. The ADE20K dataset contains more than 20K scene-centric images annotated with pixel-level objects and object parts labels. The transferability experiment results are presented in Appendix A.

**Implementation Details.** We experiment with the standard ViT supernet architecture in Tiny, Small and Base setting, *i.e.*, supernet-tiny, supernet-small and supernet-base. The decoders of the teacher and student are both implemented as a lightweight network with 2 transformer blocks. The projection network is implemented as a two-layer MLP network. The decoder networks are only used during supernet training process. After training, only the encoder is used to generate the image representation for architecture search. During supernet training, each supernet is pre-trained under a

Table 1: Performance comparison between MaskTAS and other state-of-the-art methods on ImageNet dataset, which includes three groups of models with respect to different parameter sizes, *i.e.*, tiny, small and base. Our results are highlighted in bold.

| Models | Top-1 Acc. | Top-5 Acc. | #Parameters | FLOPs | Resolution | Model Type | Design Type |
|---|---|---|---|---|---|---|---|
| MobileNetV3 Howard et al. (2019) | 75.2% | - | 5.4 M | 0.22 G | $224^2$ | CNN | Auto |
| EfficietNet-B0 Tan & Le (2019) | 77.1% | 93.3% | 5.4 M | 0.39 G | $224^2$ | CNN | Auto |
| DeiT-T Touvron et al. (2021) | 72.2% | 91.1% | 5.7 M | 1.2 G | $224^2$ | Transformer | Manual |
| AutoFormer-tiny Chen et al. (2021b) | 74.7% | 92.6% | 5.7 M | 1.3 G | $224^2$ | Transformer | Auto |
| ViTAS-Twins-T Su et al. (2022) | 79.4% | 94.8% | 13.8 M | 1.4 G | $224^2$ | Transformer | Auto |
| MaskTAS-tiny | **75.6**% | **93.1**% | **5.8** M | **1.3** G | $224^2$ | Transformer | Auto |
| ResNet50 He et al. (2016) | 79.1% | - | 25.5 M | 4.1 G | $224^2$ | CNN | Manual |
| RegNetY-4GF Radosavovic et al. (2020) | 80.0% | - | 21.4 M | 4.0 G | $224^2$ | CNN | Auto |
| EfficietNet-B4 Tan & Le (2019) | 82.9% | 95.7% | 19.3 M | 4.2 G | $380^2$ | CNN | Auto |
| BoTNet-S1-59 Srinivas et al. (2021) | 81.7% | 95.8% | 33.5 M | 7.3 G | $224^2$ | CNN + Trans | Manual |
| T2T-ViT-14 Yuan et al. (2021) | 81.7% | - | 21.5 M | 6.1 G | $224^2$ | Transformer | Manual |
| DeiT-S Touvron et al. (2021) | 79.9% | 95.0% | 22.1 M | 4.7 G | $224^2$ | Transformer | Manual |
| ViT-S/16 Dosovitskiy et al. (2020) | 78.8% | - | 22.1 M | 4.7 G | $384^2$ | Transformer | Manual |
| AutoFormer-small Chen et al. (2021b) | 81.7% | 95.7% | 22.9 M | 5.1 G | $224^2$ | Transformer | Auto |
| ViTAS-Twins-S Su et al. (2022) | 82.0% | 95.7% | 30.5 M | 3.0 G | $224^2$ | Transformer | Auto |
| MaskTAS-small | **82.5**% | **95.9**% | **22.1** M | **4.9** G | $224^2$ | Transformer | Auto |
| ResNet152 He et al. (2016) | 80.8% | - | 60 M | 11 G | $224^2$ | CNN | Manual |
| EfficietNet-B7 Tan & Le (2019) | 84.3% | 97.0% | 66 M | 37 G | $600^2$ | CNN | Auto |
| ViT-B/16 Dosovitskiy et al. (2020) | 79.7% | - | 86 M | 18 G | $384^2$ | Transformer | Manual |
| DeiT-B Touvron et al. (2021) | 81.8% | 95.6% | 86 M | 18 G | $224^2$ | Transformer | Manual |
| AutoFormer-base Chen et al. (2021b) | 82.4% | 95.7% | 54 M | 11 G | $224^2$ | Transformer | Auto |
| ViTAS-Twins-B Su et al. (2022) | 83.5% | 96.5% | 124.8 M | 16.1 G | $224^2$ | Transformer | Auto |
| MaskTAS-base | **83.8**% | **96.4**% | **53.7** M | **11** G | $224^2$ | Transformer | Auto |

100-epoch schedule on ImageNet-1K training set. To avoid the high overhead of teacher pre-training, we directly employ the MIM pre-trained models released from the official MAE implementations as our teacher model. We adopt a cosine decay schedule with a warm-up for 20 epochs. We adopt Adam optimizer with a weight decay of 0.05. The size of input image is set to 224×224 and the masking ratio is set to 90% by default. During architecture search, we employ the ImageNet validation set for model testing. We perform evolutionary search for 20 epochs to get the optimal architecture, where the population size $N_p$ is set to 50. For model fine-tuning, we also use Adam optimizer with weight decay of 0.05. We fine-tune the searched architecture for 100 epochs with a batch size of 2048, a learning rate of 5e-3, and a drop path rate of 0.1. For ADE20K semantic segmentation, we follow the same settings in MAE and adopt UperNet as our framework.

## 3.2 MAIN RESULTS ON IMAGENET

We first conduct experiments on the widely used ImageNet-1K dataset. The self-supervised architecture search process is performed over each supernet configuration, including supernet-tiny, supernet-small and supernet-base. For each configuration, we compare MaskTAS with both supervised counterparts and handcrafted architectures. As shown in Table 1, MaskTAS model families can achieve 75.6%, 82.5% and 83.8% top-1 accuracy, significantly outperforming the handcrafted CNN and transformer architectures, *e.g.*, ResNet, ViT and DeiT. This phenomenon indicates the effectiveness of automatic neural architecture search in designing superior architectures. In addition to the handcrafted architectures, we can observe from Table 1 that MaskTAS can consistently outperform the supervised transformer architecture search counterpart AutoFormer. This indicates the effectiveness of masked image modeling in self-supervised architecture search. When pre-trained for 100 epochs, MaskTAS-base outperforms AutoFormer-base pre-trained for 800 epochs by 1.4% and 0.7% on top-1 and top-5 accuracy respectively. When compared with another TAS method ViTAS-Twins, our method can achieve comparable or better results without the need for data labelling in most cases. Moreover, both the parameter size and FLOPs of MaskTAS are much less than ViTAS-Twins. The efficiency and effectiveness of MaskTAS mainly lie in two facts: 1) the MIM-based pixel reconstruction objective helps each subnet architecture to learn better local image features; 2) the distillation objective facilitates the efficient training of diverse subnet architectures.

## 3.3 ABLATION STUDIES

In this subsection, we conduct ablation studies to explore the effect of masking ratio and compare the training efficiency between MaskTAS and AutoFormer.

**Effect of Masking Ratio.** The architecture training and searching are both performed in MIM-based self-supervised paradigm, which learns to reason about the missing patches masked by a specific ratio. In order to study the effect of masking ratio on performance, we present the top-1 accuracy of MaskTAS-small with respect to different masking ratios in Figure 3. Recall that the best masking ratio of the original MAE method can be surprisingly high as 75%. From Figure 3, we can observe that the proposed MaskTAS can maintain stable performance over a wide range of masking ratios, thus allowing for a much higher ratio than MAE. In particular, compared with the 75% masking ratio of MAE, it's interesting to note that MaskTAS can raise the masking ratio to 90% without performance drop. We believe this is

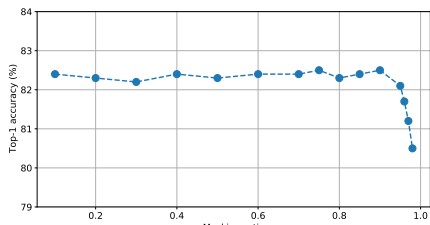

Figure 3: Top-1 accuracy performance of MaskTAS-small with respect to different masking ratios.

because the integration of knowledge distillation strategy into the masked image modelling process. With the assistance of distilled knowledge from the teacher model, the student model can make better use of visible patches, even at a very limited amount.

**Comparison of Supernet Training.** To demonstrate the training efficiency of the proposed method, we compare the supernet training process between AutoFormer and MaskTAS in Figure 4. As shown in Figure 4(a), even with the supervision of labelled data, the supernet training loss of AutoFormer converges slowly, which can't reach convergence even after 500 epochs of training. This is because the supernet contains various candidates that need to be fully optimized. In contrast, MaskTAS performs self-supervised architecture search through Masked image modeling (MIM). MIM has shown great promise for self-supervised learning yet been criticized for learning inefficiency, which further increases the difficulty of supernet training. Even with this challenge, the proposed MaskTAS can still converge to a stable value in only 100 epochs as indicated in 4(b), which significantly exceeds AutoFormer in training efficiency. We believe this is benefiting from the knowledge distillation strategy, which can provide strong supervision to allow for efficient supernet training.

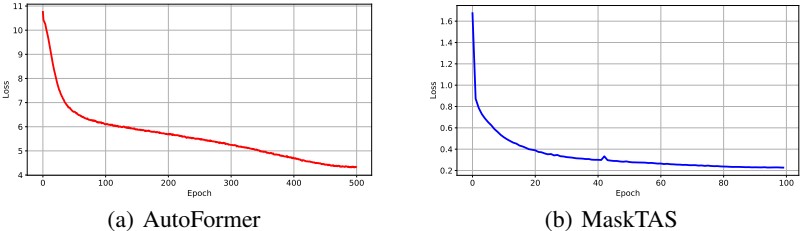

(a) AutoFormer      (b) MaskTAS

Figure 4: Comparison of supernet training between AutoFormer and MaskTAS.

## 4 CONCLUSION

In conclusion, this paper has proposed a novel self-supervised neural architecture search method, MaskTAS, that uses masked image modelling to search for efficient transformer architectures. By eliminating the need for data labeling, MaskTAS greatly reduces the expensive costs of supervised learning and enables efficient training of transformer supernet. The siamesed teacher-student architecture and the unsupervised evaluation metric based evolutionary search algorithm further enhance the learning efficiency and accuracy of the search process. The experimental results on CIFAR-10, CIFAR-100, and ImageNet datasets demonstrate that MaskTAS can achieve state-of-the-art accuracy without using manual labels. Furthermore, MaskTAS can generalize well to various data domains by searching specialized transformer architectures in a self-supervised manner. The proposed method has great potential in automating the neural architecture design process for real-world applications, saving tremendous labor from human experts, and reducing the costs of data labeling.

## ACKNOWLEDGMENTS

This work was supported by National Science and Technology Major Project (No. 2022ZD0117103), National Nature Science Foundation of China (No. 62302384, No. 62172326, No. 62192781, No. 62272374, and No. 62137002), the MOE Innovation Research Team No. IRT17R86, China University Innovation Fund No. 2021FNA04003, China Postdoctoral Science Foundation under Grant 2023M742790, Fundamental Research Funds for the Central Universities under Grant xpt012023022, and the Project of China Knowledge Centre for Engineering Science and Technology.

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

## A  TRANSFER LEARNING EXPERIMENTS

To further evaluate the searched transformer architectures of our method, we present transfer learning experiments on other popular downstream tasks.

**Transfer to Other Classification Tasks** In addition to ImageNet-1k, we also perform architecture search on other classification tasks, including CIFAR-10 Krizhevsky et al. (2009), CIFAR-100 Krizhevsky et al. (2009), PETS Parkhi et al. (2012) and Flowers Nilsback & Zisserman (2008). Based on the MaskTAS supernet pre-trained on ImageNet, we perform architecture search on each data domain to obtain the optimal architecture respectively. Then we inherit the parameter weights of each architecture from the well-optimized supernet, and fine-tune them on the corresponding dataset. The classification results of MaskTAS-small are reported in Table 2. For the purpose of fair comparison, we also present the results achieved by ViT-B and AutoFormer-S pre-trained on ImageNet-1k simultaneously. It can be easily observed from Table 2 that the proposed MaskTAS gets consistent improvement on the four datasets compared with the other two methods. We believe this improvement benefits from the self-supervised architecture search framework. Compared with the supervised counterparts, the searched architectures of MaskTAS are not limited by the range and quality of data annotation. Thus, the proposed MaskTAS is capable of producing transformer architectures that can better transfer to other data domains.

**Results on Semantic Segmentation Task** We also evaluate the generalization ability of MaskTAS by transferring the searched architectures to semantic segmentation task. Each searched architecture

Table 2: Transferability comparison on different data domains.

| Method | CIFAR-10 | CIFAR-100 | PETS | Flowers |
|---|---|---|---|---|
| ViT-B | 98.1% | 87.1% | 93.8% | 89.5% |
| AutoFormer-S | 98.3% | 88.1% | 94.1% | 97.8% |
| MaskTAS-small | **99.2**% | **90.3**% | **94.8**% | **98.6**% |

Table 3: Performance comparison (mIoU) on ADE20K semantic segmentation.

| Method | Pre-train data | Small | Base |
|---|---|---|---|
| Naive supervised | IN1K w/ labels | - | 47.4% |
| Twins-PCPVT | IN1K w/ labels | 46.2% | 47.1% |
| Twin-SVT | IN1K w/ labels | 45.9% | 47.7% |
| ViTAS-Twins | IN1K w/ labels | 47.9% | 50.2% |
| MaskTAS | IN1K w/o labels | **48.3**% | **51.6**% |

inherits the weights from the ImageNet-pre-trained supernet, and then fine-tuned on ADE20K for semantic segmentation. The experimental results of different transformer models in small and base size are reported in Table 3. More specifically, we compare five methods in Table 3: Naive supervised, Twins-PCPVT Chu et al. (2021), Twin-SVT Chu et al. (2021), ViTAS-Twins Su et al. (2022) and the proposed MaskTAS. Naive supervised indicates the supervised pre-training method done from scratch, in which we directly use the reported mIoU from He et al. (2022). Twins-PCPVT Chu et al. (2021), Twin-SVT Chu et al. (2021) and ViTAS-Twins Su et al. (2022) are the other three supervised counterparts. Without the access of manual labels, the proposed MaskTAS is at a disadvantage in classification than the supervised counterparts. Even so, it is noteworthy that MaskTAS can still achieve better performance than the other methods. In particular, the small and base model of MaskTAS can outperform the second best model by 0.4% and 1.4% respectively, showing that the self-supervised pre-training can benefit dense downstream tasks as well.

