# OpenReview forum: "Masked Distillation Advances Self-Supervised Transformer Architecture Search"
_ICLR.cc/2024/Conference — ICLR 2024 poster_

### Official Review · Reviewer_nfTy · 2023-10-30

**Soundness:** 3 good
**Presentation:** 3 good
**Contribution:** 3 good
**Rating:** 8
**Confidence:** 2

**Summary:**

In this paper, the authors propose a masked image modelling (MIM) based self-supervised neural architecture search method specifically designed for vision transformers. The paper avoids the expensive costs of data labeling inherited from supervised learning. The paper can generalize well to various data domains and tasks by searching specialized transformer architectures in self-supervised manner.

**Strengths:**

1. This paper is written in a clear and accessible manner, making it easy to comprehend.
2. The utilization of mask image modeling for the search of efficient Transformer structures significantly mitigates the high costs associated with supervised learning
3. This paper formulates the configuration of MaskTAS's search space as resembling a teacher-student framework, and the notion of extracting knowledge from pre-trained networks is indeed a noteworthy proposition.

**Weaknesses:**

1. The authors mentioned that experiments were conducted on CIFAR-10 and CIFAR-100 for assessing the transferability of self-supervised architecture search methods. However, it should be noted that the paper does not provide the corresponding experimental results.
2. In Table 1, it is evident that the approach introduced in this paper underperforms when compared to the use of a CNN model, across multiple dimensions, including performance, parameter size, and FLOPs. What is the rationale behind the author's decision to employ the Transformer architecture?

**Questions:**

Please refer to the weaknesses.

---

> ### Author Response · Authors · 2023-11-20
>
> We thank the reviewer for the valuable feedback and thoughtful comments. We appreciate the reviewer's acknowledgment that our work mitigates the high costs associated with supervised learning, formulates a noteworthy proposition of extracting knowledge from pre-trained networks, and the paper is written in a clear and accessible manner. In the following, we provide detailed explanations to address the concerns.
> ***
> **Q1: The experiments conducted on CIFAR-10 and CIFAR-100 for assessing the transferability of self-supervised architecture search methods.**
>
> We apologize for missing the results on CIFAR-10 and CIFAR-100 in the manuscript. We have provided the corresponding results below in Table 1 and added these results to the current version of our paper. For the purpose of comparison, we also present the results achieved by ViT-B and AutoFormer-S pre-trained on ImageNet-1k in Table 1. It can be observed that the proposed MaskTAS gets consistent improvement on the four datasets compared with ViT-B and AutoFormer-S. We believe this improvement benefits from the self-supervised architecture search framework. Compared with the supervised counterparts, the searched architectures of MaskTAS are not limited by the range and quality of data annotation. Thus, the proposed MaskTAS is capable of producing transformer architectures that can better transfer to other data domains.
>
> Method | CIFAR-10 | CIFAR-100 | PETS | Flowers
> -------------------|------------------|------------------|------------------|------------------
> ViT-B | 98.1\% | 87.1\% | 93.8\% | 89.5\%
> AutoFormer-S | 98.3\%  | 88.1\%  | 94.1\%  | 97.8\%
> MaskTAS-small | **99.2\%** | **90.3\%** | **94.8\%** | **98.6\%**
>
> **Table 1** Transferability comparison on different data domains.
>
> ***
>
> **Q2: What is the rationale behind the author's decision to employ the Transformer architecture?**
>
> From Table 1 of the paper, we can observe that MaskTAS is competitive compared to vanilla CNN models. Moreover, our MaskTAS also performs better than the manually-designed ResNet [1], ResNeXt [2] and DenseNet [3], demonstrating the potentials of pure transformer models for visual representation. However, transformer-based vision models now are still inferior to the lightweight CNN models based on inverted residual blocks, such as MobileNetV3 [4] and EfficientNet [5]. The reason is that inverted residuals are optimized for edge devices, so the model sizes and FLOPs are much smaller than vision transformers. Motivated by this, we plan to investigate the efficient search and deployment of ViT architectures on mobile devices in forthcoming research. This requires to optimize the design choices of ViT architecture to adapt to mobile devices, and develop new transformer architecture search method with low latency and high parameter efficiency. Your feedback has been valuable in shaping the direction of our future investigations.
>
> ***
> **References:**
>
> [1] He K, Zhang X, Ren S, et al. Deep residual learning for image recognition[C]//Proceedings of the IEEE conference on computer vision and pattern recognition. 2016: 770-778.
> [2] Xie S, Girshick R, Dollár P, et al. Aggregated residual transformations for deep neural networks[C]//Proceedings of the IEEE conference on computer vision and pattern recognition. 2017: 1492-1500.
> [3] Huang G, Liu Z, Van Der Maaten L, et al. Densely connected convolutional networks[C]//Proceedings of the IEEE conference on computer vision and pattern recognition. 2017: 4700-4708.
> [4] Howard A, Sandler M, Chu G, et al. Searching for mobilenetv3[C]//Proceedings of the IEEE/CVF international conference on computer vision. 2019: 1314-1324.
> [5] Tan M, Le Q. Efficientnet: Rethinking model scaling for convolutional neural networks[C]//International conference on machine learning. PMLR, 2019: 6105-6114.

---

> > ### Comment · Reviewer_nfTy · 2023-11-21
> > **Good**
> >
> > Thanks very much for your rebuttal.

---

### Official Review · Reviewer_42Bt · 2023-10-31

**Soundness:** 4 excellent
**Presentation:** 4 excellent
**Contribution:** 3 good
**Rating:** 8
**Confidence:** 4

**Summary:**

This paper presents a self-supervised NAS method for vision transformers (ViT). Specifically, the proposed method applied masked image modeling as the self-supervised method. The transformer supernet is composed of two models, a fixed-arch teacher model and a student super-network which were optimized simultaneously. The knowledge distillation strategy was used to allow for efficient self-supervised training of the supernet, and an unsupervised evolutionary search algorithm was designed to achieve self-supervised transformer architecture search. The paper is well written and easy to understand. The authors have conducted quite extensive experiments which show that the proposed method achieves good results in general. In particular, the searched architectures can achieve state-of-the-art accuracy on benchmark dataset without using manual labels.

**Strengths:**

+ The proposed method designs the first self-supervised NAS technique for vision transformers (ViT). The shining point is that It proposes the first unsupervised evolutionary search algorithm, in order to achieve self-supervised transformer architecture search.
+ The experiments are quite solid and demonstrate SOTA performance without using manual labels.
+ The paper is well written. The motivation and problem statement are clear and easy to understand.

**Weaknesses:**

- I was not able to find the architecture of the final searched model in the paper. I would like to see the entire architecture of searched models, rather than a single number that indicating the final performance. This is particularly important in the NAS literature.
- Typically, in a NAS paper, the final output is the architecture, which should be trained again from scratch to test the "true" performance. In this work, the weights that are distilled during training are preserved and used for fine-tuning in the downstream tasks. It would be better if the authors could provide an explanation on this.
- Some of the experimental details are missing, for instance, where do all the numbers in Table 1 come from?

**Questions:**

Please refer to above.

---

> ### Author Response · Authors · 2023-11-20
>
> We are grateful for your insightful comments and constructive suggestions. We provide a thorough response to each point raised to ensure clarity and a comprehensive understanding of our approach.
> ***
> **Q1: The architecture of the final searched model.**
>
> Thanks for the valuable suggestion. We provide the searched architectures on ImageNet-1k as follows. In order to present the network architecture more clearly, we will present them in the form of diagrams and add them to the supplementary material.
>
> MaskTAS-tiny: \{*layer num*: 12, *embed dim*: 192, *mlp ratio*: [4.0, 4.0, 4.0, 4.0, 4.0, 3.5, 3.5, 4.0, 4.0, 3.5, 4.0, 4.0], *head num*: [4, 4, 4, 4, 3, 3, 3, 3, 3, 4, 3, 4]\}
> MaskTAS-small: \{*layer num*: 13, *embed dim*: 384, *mlp ratio*: [4.0, 4.0, 3.0, 4.0, 3.0, 4.0, 3.0, 3.0, 4.0, 3.0, 3.0, 4.0, 3.0]; *head num*: [7, 7, 7, 7, 5, 7, 7, 5, 6, 5, 6, 6, 5]\}
> MaskTAS-base: \{*layer num*: 14, *embed dim*: 576, *mlp ratio*: [3.5, 4.0, 3.0, 4.0, 4.0, 3.5, 3.0, 3.5, 3.0, 3.0, 3.0, 3.0, 3.0, 3.0]; *head num*: [10, 9, 10, 10, 10, 10, 9, 10, 9, 9, 10, 9, 9, 10]\}
>
> ***
> **Q2: The architectures should be trained again from scratch to test the true performance.**
>
> For one-shot NAS, pre-training a high-quality supernet is essential for candidate architectures to inherit weights directly [1,2]. Thus, in addition to training from scratch, inheriting the pre-trained weights for fine-tuning is also a popular way of architecture retraining. For better comparison, we provide the results of both fine-tuning and training from scratch in Table 1. The results on all three specifications of model size, *i.e.*, tiny, small and base, show that our MaskTAS consistently outperforms AutoFormer in both fine-tuning and training from scratch settings.
>
> Method | Model Size | Finetune | Train from scratch
> -------------------|------------------|------------------|------------------
> AutoFormer-T | 5.7M | 74.9\% | 74.9\%
> MaskTAS-tiny | 5.5M | 75.6\% | 75.3\%
> AutoFormer-S | 22.9M | 81.8\% | 81.7\%
> MaskTAS-small | 22.1M | 82.5\% | 82.0\%
> AutoFormer-B | 53.7M | 82.6\% | 82.6\%
> MaskTAS-base | 52.0M | 83.8\% | 82.6\%
>
> **Table 1** Top-1 accuracy comparison achieved by fine-tuning and training from scratch.
>
> ***
> **Q3: Some of the experimental details are missing, for instance, where do all the numbers in Table 1 come from?**
>
> We are sorry for the missing experimental details. In Table 1 of our paper, we compare the performance of the proposed MaskTAS with existing visual models. Specifically, the performance comparison with existing transformer models utilizes the results reported by AutoFormer [1] and ViTAS [3]. The results of CNN models in the table are mostly inherited from the corresponding paper. In addition, we will carefully review the experimental part and try to complete the experimental details.
>
> ***
> **References:**
>
> [1] Chen M, Peng H, Fu J, et al. Autoformer: Searching transformers for visual recognition[C]//ICCV 2021: 12270-12280.
> [2] Yu J, Jin P, Liu H, et al. Bignas: Scaling up neural architecture search with big single-stage models[C]//ECCV 2020: 702-717.
> [3] Su X, You S, Xie J, et al. ViTAS: Vision transformer architecture search[C]//ECCV 2022: 139-157.

---

> > ### Comment · Reviewer_42Bt · 2023-11-22
> > **Thanks for the Rebuttal**
> >
> > Thanks for the rebuttal, which addresses the concerns from my end. I would like to increase my score.

---

### Official Review · Reviewer_YXCu · 2023-10-31

**Soundness:** 3 good
**Presentation:** 3 good
**Contribution:** 3 good
**Rating:** 6
**Confidence:** 5

**Summary:**

This paper proposes to combine masked image model with transformer architecture search in a self-supervised setting. The author designed a new metric to measure the searched results from the supernet. The proposed method achieves competitive results on ImageNet.

**Strengths:**

The experiments are comprehensive, including imagenet/cifar/ade20k.
The writing is good, the motivation is clear, and the method is sound.

**Weaknesses:**

The only concern is the unfair comparison in the experiments. It is unclear whether the performance gain is coming from knowledge distillation or training with the reconstruction objective.

Therefore, I suggest the author report the experimental results using only masked image modeling on ImageNet & Cifar & ADE20K, since the baseline that MaskTAS compares did not use KD in their approach.

**Questions:**

I will raise my rating if the results using only masked image modeling can outperform SOTA methods.

---

> ### Author Response · Authors · 2023-11-20
>
> We thank the reviewer for the valuable feedback and thoughtful comments. We appreciate that the reviewer acknowledges the writing is good, the motivation is clear, and the method is sound. We will address your concerns in detail and we hope our response can further demonstrate the strengthens of our method.
> ***
> **Q1: It is unclear whether the performance gain is coming from knowledge distillation or training with the reconstruction objective.**
>
> Thanks for the valuable suggestion. We have conducted ablation studies to explore the impact of self-supervised learning and teacher-student distillation individually. The results are reported in Table 1.
> Method | w/ $SSL$&$KD$ |	w/o $SSL$ |	w/o $KD_{100}$ |	w/o $KD_{1600}$
> -------------------|------------------|------------------|------------------|------------------
> MaskTAS-tiny | 75.6\% | 75.0\% | 73.3\% | 75.5\%
> MaskTAS-small | 82.5\% | 81.8\% | 80.6\% | 82.5\%
> MaskTAS-base | 83.8\% | 82.7\% | 81.2\% | 84.0\%
>
> **Table 1**  Ablation study on the impact of self-supervised learning and teacher-student distillation.
>
> From Table 1, we can make the following observations. 1) The model equipped with both the two components, i.e., ''w/ $SSL$&$KD$'', surpasses that with only distillation, i.e., ''w/o $SSL$''. We think this is because of the capacity gap between teacher and student, thus the distillation process may cause information losses. 2) For better comparison, we adopt two configurations for the model without distillation, where ''w/o $KD_{100}$'' and ''w/o $KD_{1600}$'' refer to the model with 100 and 1600 pre-training epochs respectively. Although ''w/o $KD_{100}$'' suffers from a significant performance degradation, ''w/o $KD_{1600}$'' achieves comparable performance as ''w/ $SSL$&$KD$'' by increasing the number of pre-training epochs. 3) In conclusion, self-supervised learning is beneficial for performance improvement, while distillation helps speed up training. Thus, our MaskTAS employs both of the two components to achieve a tradeoff between performance and overhead.
> ***
> **Q2: The experimental results using only masked image modeling on ImageNet & Cifar & ADE20K.**
>
> Thanks for the suggestion. The results using only masked image modeling on ImageNet have been provided in Table 1. We further report the results on CIFAR & ADE20K in Table 2, where ''w/o $KD_{100}$'' and ''w/o $KD_{1600}$'' pre-train the model using only masked image modeling with 100 and 1600 epochs respectively.
> Method | CIFAR-10 (Top-1 acc) | CIFAR-100 (Top-1 acc) | ADE20K (mIoU)
> -------------------|------------------|------------------|------------------
> MaskTAS-small | 99.2\% | 90.3\% | 48.3\%
> MaskTAS-small w/o $KD_{100}$  | 98.3\% | 88.7\% | 46.5\%
> MaskTAS-small w/o $KD_{1600}$ | 99.1\% | 90.5\% | 48.4\%
>
> **Table 2** Performance using only MIM on CIFAR and ADE20K.
>
> Notably, by increasing the pre-training epochs from 100 to 1600, ''MaskTAS-small w/o KD'' can achieve comparable or even better performance than the complete ''MaskTAS-small'' model, as shown in Table 2. It indicates the great potential of masked image modelling in self-supervised architecture search.

---

> ### Comment · Reviewer_YXCu · 2023-12-04
> **Comments**
>
> After carefully reading the authors' responses and other reviewers' comments, I raised my rating to 6. Although I maintain some reservations about the experimental approach, particularly as the new experiments suggest that the performance gains are primarily attributable to knowledge distillation, I think this paper meets the bar of ICLR.

---

### Official Review · Reviewer_zgPr · 2023-11-01

**Soundness:** 3 good
**Presentation:** 3 good
**Contribution:** 3 good
**Rating:** 6
**Confidence:** 4

**Summary:**

This paper incorporates the MIM based self-supervised learning with the NAS framework for vision transformers. It also introduces a siamesed teacher-student architecture for the NAS search space to distill knowledge from pre-trained networks, which shows better convergence.
It also develops a novel unsupervised evaluation metric for the evolutionary search algorithm, where teacher networks guide the student branch search.
Experimental results show competitive results to STOA accuracy supervised and unsupervised model results.

**Strengths:**

1. This paper introduces the self-supervised learning framework into the NAS framework for vision transformers.
2. This paper also introduces a siamesed teacher-student architecture to facilitate the supernet training convergence.
2. Experimental results show competitive results to STOA accuracy supervised and unsupervised model results.

**Weaknesses:**

1. There are some statements in the paper which needs further clarification (see details in the Question section)
2. It's not clear the impact of the self-supervised learning and teach-student distillation framework individually. Is it possible to add some ablation studies?
3. It's not clear on the overhead and tradeoff of training another teacher encoder-decoder network.

**Questions:**

1. Could you clarify how does MaskTAS "which completely avoids the expensive costs of data labeling inherited from supervised learning"? How does MaskTAS perform classification tasks? Will labels be need for fine-tuning the classification heads?
2. Is there any evidence to support MaskTAS can "generalize well to various data domains and tasks by searching specialized transformer architectures in self-supervised manner"?

---

> ### Author Response · Authors · 2023-11-20
>
> We greatly appreciate your careful reading and constructive comments. We have carefully considered each comment and made revisions accordingly.
> ***
> **Q1-1: MaskTAS avoids the expensive costs of data labeling.**
>
> As an architecture search method for vision transformers, MaskTAS is mainly designed to avoid the data labeling cost for architecture search. By developing self-supervised architecture search framework, MaskTAS can discover optimal transformer architectures without using manual labels. Then, these searched architectures can be adapted to downstream tasks through a fine-tuning or re-training process, where different tasks may have different requirements for data labels. In terms of the standard classification task, labels are needed for fine-tuning the classification heads.
> ***
> **Q1-2: MaskTAS generalizes well to various data domains and tasks.**
>
> We thank the reviewer for the valuable comment. We provide evidences to support the generalization ability of MaskTAS. 1) In addition to ImageNet-1k, we also perform architecture search on other data domains, including CIFAR-10, CIFAR-100, PETS and Flowers. The classification results are reported in Table 1. 2) We also evaluate the generalization ability of MaskTAS by transferring the searched architectures to semantic segmentation task. The experimental results of searched transformer models in small and base size are reported in Table 2. We have added these results and corresponding analyses to our paper.
> Method | CIFAR-10 | CIFAR-100 | PETS | Flowers
> -------------------|------------------|------------------|------------------|------------------
> ViT-B | 98.1\% | 87.1\% | 93.8\% | 89.5\%
> AutoFormer-S | 98.3\%  | 88.1\%  | 94.1\%  | 97.8\%
> MaskTAS-small | **99.2\%** | **90.3\%** | **94.8\%** | **98.6\%**
>
> **Table 1** Transferability comparison on different data domains.
>
> Method | Pre-train data | Small | Base
> -------------------|------------------|------------------|------------------
> Naive supervised | IN1K w/ labels | - | 47.4\%
> Twins-PCPVT | IN1K w/ labels | 46.2\% | 47.1\%
> Twin-SVT | IN1K w/ labels | 45.9\% | 47.7\%
> ViTAS-Twins | IN1K w/ labels | 47.9\% | 50.2\%
> MaskTAS | IN1K w/o labels | **48.3\%** | **51.6\%**
>
> **Table 2** Performance comparison (mIoU) on ADE20K semantic segmentation.
> ***
> **Q2: The impact of the self-supervised learning and distillation individually.**
>
> Thanks for the valuable suggestion. We have conducted ablation studies to explore the impact of self-supervised learning and teacher-student distillation individually. The results are reported in Table 3.
> Method | w/ $SSL$&$KD$ | w/o $SSL$ | w/o $KD_{100}$ | w/o $KD_{1600}$
> -------------------|------------------|------------------|------------------|------------------
> MaskTAS-tiny | 75.6\% | 75.0\% | 73.3\% | 75.5\%
> MaskTAS-small | 82.5\% | 81.8\% | 80.6\% | 82.5\%
> MaskTAS-base | 83.8\% | 82.7\% | 81.2\% | 84.0\%
>
> **Table 3**  Ablation study on the impact of self-supervised learning and knowledge distillation.
>
> From Table 3, we can make the following observations. 1) The model equipped with both the two components, *i.e.*, ''w/ $SSL$&$KD$'', surpasses that with only distillation, *i.e.*, ''w/o $SSL$''. We think this is because of the capacity gap between teacher and student, thus the distillation process may cause information losses. 2) For better comparison, we adopt two configurations for the model without distillation, where ''w/o $KD_{100}$'' and ''w/o $KD_{1600}$'' refer to the model with 100 and 1600 pre-training epochs respectively. Although ''w/o $KD_{100}$'' suffers from a significant performance degradation, ''w/o $KD_{1600}$'' achieves comparable performance as ''w/ $SSL$&$KD$'' by increasing the number of pre-training epochs. 3) In conclusion, self-supervised learning is beneficial for performance improvement, while distillation helps speed up training. Thus, our MaskTAS employs both of the two components to achieve a tradeoff between performance and overhead.
> ***
> **Q3: The overhead and tradeoff of training another teacher encoder-decoder network.**
>
> Due to the MIM-based self-supervised learning strategy, pre-training another teacher encoder-decoder network can inevitably bring a high computational overhead, *e.g.*, MAE [1] and its variants [2-3]. In our work, we directly employ the MIM pre-trained models released from the official MAE implementations as our teacher model to avoid the high overhead of teacher pre-training. Surprisingly, we find by building upon the teacher pre-trained in MAE, the whole distillation framework can efficiently yield high-performance student models with performance and overhead trade-offs.
> ***
> **References:**
>
> [1] He K, Chen X, Xie S, et al. Masked autoencoders are scalable vision learners[C]//CVPR 2022.
> [2] Dong X, Bao J, Zhang T, et al. Bootstrapped masked autoencoders for vision BERT pretraining[C]//ECCV 2022.
> [3] Feichtenhofer C, Li Y, He K. Masked autoencoders as spatiotemporal learners[J]//NeurIPS 2022.

---

### Meta-Review · Area_Chair_eSDg · 2023-12-05

**Metareview:**

This paper presents an approach to do architecture search for transformers using self-supervised learning and knowledge distillation.
The authors show that using both these signals improves NAS. Experiments on multiple different image recognition benchmarks show that the proposed architecture improves performance.
The paper presentation and writing is quite clear.
The ablation experiments in the author response suggest that the main convergence and performance benefit come from knowledge distillation which weakens the contribution of self-supervised learning using the MIM objective.

**Justification For Why Not Higher Score:**

The ablation experiments in the author response suggest that the main convergence and performance benefit come from knowledge distillation which weakens the contribution of self-supervised learning using the MIM objective.

**Justification For Why Not Lower Score:**

A simple and technically correct way to do NAS for ViTs is useful. The execution shows solid results on multiple benchmarks.

---

### Decision · Program_Chairs · 2024-01-16

Accept (poster)